# What Awaits Myanmar's Uplands Farmers? Lessons Learned from Mainland Southeast Asia

**Martin Rudbeck Jepsen [1,\*], Matilda Palm [2] and Thilde Bech Bruun [1]**

[1] Section for Geography, Department of Geosciences and Natural Resource management, University of Copenhagen, DK-1350 Copenhagen K, Denmark; thbb@ign.ku.dk

[2] Department of Space, Earth and Environment, Physical Resource Theory, Chalmers University of technology, SE-412 96 Gothenburg, Sweden; matilda.palm@chalmers.se

\* Correspondence: mrj@geo.ku.dk; Tel.: +45-3532-2564

**Abstract:** Mainland Southeast Asia (MSA) has seen sweeping upland land use changes in the past decades, with transition from primarily subsistence shifting cultivation to annual commodity cropping. This transition holds implications for local upland communities and ecosystems. Due to its particular political regime, Myanmar is at the tail of this development. However, with Myanmar's official strategy of agricultural commercialization and intensification, recent liberalization of the national economy, and influx of multinational agricultural companies, the effects on upland land transitions could come fast. We analyze the current state of upland land use in Myanmar in a socio-economic and political context, identify the dynamics in three indicator commodity crops (maize, cassava, and rubber), and discuss the state driven economic, tenurial and policy reforms that have occurred in upland areas of mainland Southeast Asian countries in past decades. We draw on these insights to contextualize our study and hypothesize about possible transition pathways for Myanmar. The transition to annual commodity cropping is generally driven by a range of socio-economic and technical factors. We find that land use dynamics for the three indicator crops are associated with market demand and thus the opening of national Southeast-Asian economies, research and development of locally suitable high yielding varieties (HYVs), and subsidies for the promotion of seeds and inputs. In contrast, promotion of HYVs in marginal areas and without adequate agricultural extension services may results in agricultural contraction and yield dis-intensification. The environmental impacts of the transition depend on the transition pathway, e.g., through large-scale plantation projects or smallholder initiatives. The agricultural development in upland MSA follows a clear diffusion pattern with transition occurring first in Thailand, spreading to Vietnam, Cambodia and Laos. While these countries point to prospects for Myanmar, we hypothesize that changes will come slow due to Myanmar's sparse rural infrastructure, with uncertainty about tenure, in particular in areas still troubled by armed conflicts, and unwillingness of international investors to approach Myanmar given the recent setbacks to the democratization process.

**Keywords:** land transitions; commodification; land use changes; environment; Myanmar; Southeast Asia

## 1. Introduction

Land use systems and traditional livelihoods form a close-knit nexus: living off the land, rural communities adapt to and shape ecosystems through land use, and livelihoods depend on land use outcomes. In the uplands of Mainland Southeast Asia (MSA), shifting cultivation and other agroforestry practices have been the dominant livelihood activities among indigenous communities and smallholder farmers with various cultural, ethnic, and social backgrounds. Despite being linked to poverty [1–3] and environmental degradation [4], many scientists highlight that shifting cultivation

exerts moderate pressure on ecosystems compared to annual cropping and provides both staple crops and non-timber forest products to local communities [5–8]. Traditional land use systems and their users are currently in transition driven by global change processes such as increasing market integration of local land and natural resources [9]. In MSA, liberalization policies have brought development and increased well-being during the past 30 years, but have also exposed marginalized communities to new and potentially disruptive forces. The transition from traditional upland land use systems to continuous cropping systems has accelerated along multiple pathways in the past decade [10–13] with, in particular, shifting cultivation being replaced by annual cropping of maize for fodder and plantations with perennial crops such as coffee, acacia, banana, and rubber [14–16]. This commodification transition has implications for local indigenous communities, simultaneously restricting their scope and influence, but also opening new opportunities for land use activities and livelihoods [17], e.g., income stability through contract farming, increased crop production by agricultural intensification, or cash income through sales of land. At the same time, the transition can be associated with loss of land or tenure rights, increased exposure to market price fluctuations and indebtedness due to higher costs of agricultural inputs.

However, lack of reliable data and scarce research documentation of land use and transition pathways thwart a full understanding of the impacts on society and environment. Due to its particular political regime, Myanmar is at the tail of this land transition. However, with Myanmar's official strategy of agricultural commercialization and intensification [18], recent liberalization of the national economy, and influx of multinational agricultural companies, the effects on land transitions are likely to come fast. To the best of our knowledge, no research has been published on general land use and land use change in Myanmar; a single study of deforestation at national scale points to deforestation drivers [16], complemented by several local scale case studies (e.g., [19,20]).

Similarly, the impact of a commodification transition on local livelihoods and land use systems in Myanmar remain understudied and poorly understood. This is unfortunate, since recent research from other parts of Southeast Asia documents that traditional land use systems and smallholder livelihoods could be at risk when exposed to new policies, actors and market forces [21].

This paper's main objective is to analyze the current state of upland land use in a socio-economic, environmental and political context. We identify commodity-crop-related land use dynamics occurring in upland areas of mainland Southeast Asian over the past decades, discuss the drivers behind these, and draw on these experiences to contextualize our study and hypothesize possible transition pathways for Myanmar. We address this specifically through four main research questions:

1. What is the current state of traditional upland land use systems in Myanmar?
2. What have been the driving forces of land use transitions in Mainland Southeast Asia?
3. What are the key impacts related to livelihoods and the environment during recent land use transitions?
4. What are the potential pathways of future land use transitions in Myanmar and the possible impacts related to livelihoods and the environment?

We hypothesize that upland agricultural land use changes and impacts in Myanmar are or may become comparable to those seen in the remaining MSA. This is supported by comparable agro-climatic conditions, similarities in traditional farming systems (notably shifting cultivation), joint experiences with totalitarian and/or military government regimes, recent liberalizations of national economic systems, and the exposure to market drivers, megatrends promoting boom crops, contract farming, and a growing regional demand for food, fibers and energy. However, this hypothesis could fail due to Myanmar's particularities: being the Eastern border post towards India, Myanmar has direct physical access to the Indian market with different dietary preferences than MSA. This could impact crop choice and farming system in Myanmar. Furthermore, despite a ceasefire agreement, national hegemony has not been reached, and armed conflicts still occur between government military and armed rebellions.

Finally, we cannot predict if the drivers and changes seen in MSA during the past 50 years will repeat, or if new megatrends and boom crops will emerge.

## 2. Methodology

Land use, land use changes, and their economic, social, and environmental impacts, have been studied throughout MSA by academics, NGOs and donor agencies. Many academic studies are conducted and published as local-scale studies, while there is a scarcity of synthesis studies generalizing local trends and drivers and a lack of national-scale studies, and the scope of this study does not allow a full-scale systematic review or meta-analysis. A large body of knowledge is published by NGOs and donor agencies as reports, working papers, etc. Finally, Myanmar has only recently been accessible for foreign researchers, further limiting the available knowledge base. These conditions render a systematic review of general land changes and associated impacts difficult. Our methodological approach has therefore been eclectic and pragmatic (Figure 1); to assess the current state of Myanmar's upland land use systems (RQ 1), we complemented the few published peer reviewed studies with broad Google Scholar and general Internet searches for relevant reports and working papers, in addition to personal experience from field visits in Chin State. Based on these sources, we provide a summary of recent land use reforms and an overview of the traditional upland farming system, shifting cultivation. We used time series statistics from FAOSTAT on three indicator upland crops (rubber, cassava, maize) to describe expansion, contraction, intensification, and disintensification processes and relate those to drivers of change identified in the references. A similar approach was followed to identify the driving forces of land use transitions in MSA (RQ 2)), where we also included newspaper articles as a source. The key social, economic, and environmental impacts of land transitions (RQ 3) were estimated qualitatively by consulting the authors' own publications and reference databases and discussed individually for the three indicator upland crops. The results of RQs 1, 2, and 3 inform a discussion of potential pathways for future land use transitions in Myanmar (RQ 4), where we point to the possible social, economic and environmental consequences of these. The structure of the article follows this progression through the research questions.

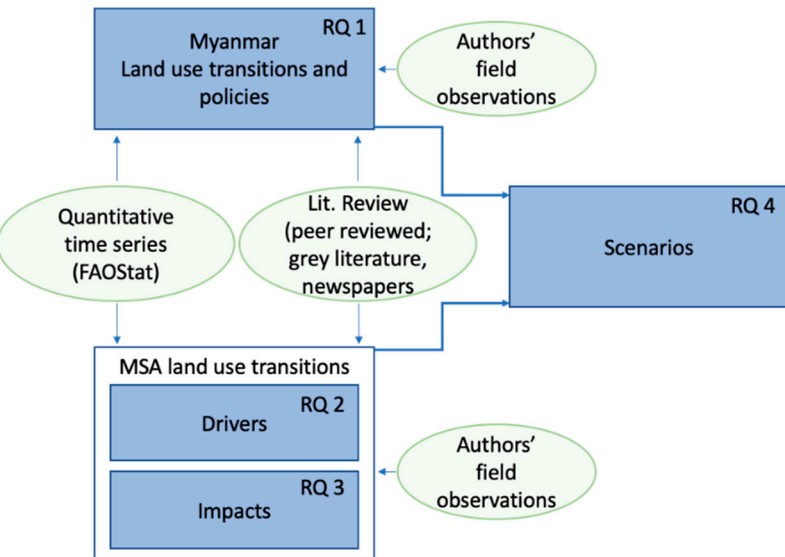

**Figure 1.** Logical flow model of the manuscript. Blue boxes refer to research questions, green ellipses to the applied methods and data sources.

## 3. Current State of Upland Land Use Systems in Myanmar

*3.1. Agricultural/Land Use Reforms in Myanmar*

70% of Myanmar's population live and work in rural areas, making agriculture a fundamental part of the society and local livelihoods [22]. Unfortunately, the agricultural sector is troubled by a strong regime maintaining control of crop choice in certain areas and weak institutions to grant, monitor, and enforce land rights.

Historically, land rights in Myanmar have been recognized as customary rights/village land/communal land or state land. During the military regime (1962–2011), the land was nationalized and strict control on crop choices and marketing of produce was enforced. In 1987, the military rule loosened its tight grip on the agricultural sector, allowed farmers to decide on which crops to grow (except in designated paddy rice production areas), to sell their crops on the domestic and export markets, and lifted government price control [23]. This also opened the way for shady land acquisitions; it is estimated that approximately 0,77 Mha were illegally transferred to private companies between 1993 and 2013, even though 70 percent of that land has never been developed and is still farmed by the original users [24].

Two land laws (the Farmland Law and the Vacant, Fallow and Virgin Land Law), passed in 2012, significantly changed the governance of land. The two laws stipulate that land can be bought, sold and transferred on a land market with land use certificates. In a country where large numbers of smallholders lack formal land titles this is highly problematic [25]. The laws allow the central government to re-allocate smallholders' farm and forest lands—both upland shifting cultivation land, especially fallows, and lowlands without official land title—to domestic and foreign investors. The laws do not take into account the land rights of ethnic minorities and fail to recognize customary and communal tenure systems.

As a result, upland smallholders are under threat of losing their lands, and many have already been targeted for resource extraction, agribusiness concessions, and mega infrastructure projects [26]. By mid-2013, the government had allocated as much as 2 Mha to agribusiness concessions [27], and 5 Mha in Myanmar (7.5% of the total land area) is reportedly designated to concessions on current 'wasteland' financed by foreign and joint venture investors [28,29]. In addition, domestic companies had been granted almost 1 Mha of 'fallow and virgin land' for commercial agricultural production. This development is in line with the laws from 2012, which ultimately seek to convert 'idle' land into 'modern agriculture', with a preference for large-scale (foreign) investments [30]. Both laws are mainly benefitting commercial interest and have facilitated land grabbing and created several land related conflicts with increasing protests by local communities affected by these developments [31].

In response to growing criticism, the Government drafted a new National Land Use Policy (NLUP) in 2016 [32], recognizing customary and communal land rights, but so far without institutions to identify, verify, and secure land rights. It thus remains to be seen how well authorities comply with the policy intentions, embrace them in law acts, build strong institutions to enforce the laws, and hence protect vulnerable smallholders from land grabs and forced evictions. However, the 2017 draft Land Acquisition Act does not recognize the rights of non-title land holders, and only vaguely for customary and communal land rights often practiced in ethnic groups [33], placing the NULP as mere greenwashing.

*3.2. Contract Farming Scheme Example*

A manifestation of the concessions granted is the contract farming scheme implemented in the early 1990s by Thai Charoen Pokphand Group (CP) in upland rural Shan State as a part of a US-supported opium substitution program. Since then, the uplands of Shan State have developed into one of CP's important maize production zones. In the mid-2000s, the governments of Thailand and Myanmar signed a bilateral Memorandum of Understanding to set aside 700,000 ha of "vacant and fallow land" for CP maize contract farming. However, the implementation of this agreement

stalled - partly due to political impasse with the armed Karen groups in Shan state. Nevertheless, CP engagement in contract farming has triggered a significant agrarian transformation from traditional upland land use systems to cash cropping of high input hybrid maize. According to CP, about 300,000 ha is currently used for production of industrial maize and CP is trying to expand production of maize in Karen State along the Thai border for the Thai domestic chicken feed market. These attempts began after the ceasefire agreement with the Karen armed group [34].

### 3.3. Traditional Upland Land Use Systems

In this section, we provide a general description of traditional upland agriculture in Myanmar. We define 'upland' as lands with slopes steeper than 15 degrees, where agriculture is thought to be rainfed or to rely on rainfed irrigation, not to be prone to flooding and not to be suitable for large-scale mechanization. These areas comprise 27.5% of the total area of Myanmar (Figure 2). This definition excludes, e.g., high plains, i.e., relatively level lands at higher altitudes, because these are suitable for mechanized agriculture and thus are not representative of traditional smallholder agriculture. This delimitation implies that we ignore the status of Myanmar's plains, mainly used for paddy rice and pulses, as well as the land use dynamics in the Southern Tanintharyi region, where oil palm plantations are emerging [35].

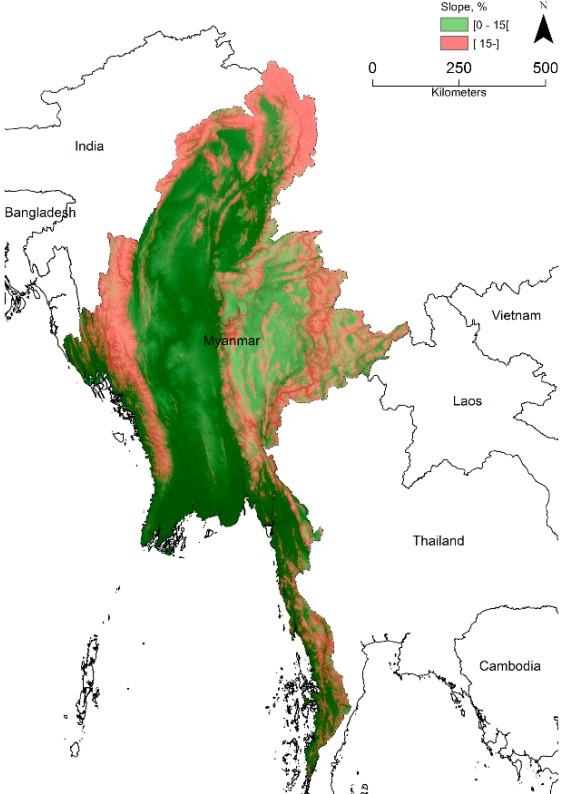

**Figure 2.** Upland areas of Myanmar defined as slopes > 15 degrees are displayed in red. Slopes < 15 are green. The slope map is overlaid on a digital elevation model which dictates the shading of red and green colors.

Inhabitants of the upland areas of Myanmar have traditionally relied on shifting cultivation, locally known as '(shwe pyaung) taungya', which means 'moving hill-farm'. Shifting cultivation is the dominant land use system in Kachin, Kayah, Kayin, Chin and Shan states, but is also practiced in other states and regions of Myanmar [36,37].

Shifting cultivation is a land use system that involves rotation of fields, rather than crops, and relies on the use of fallow to restore land productivity [38]. The fallows are cleared by means of slashing and

burning, the land is cropped with subsistence crops for a short period of time, and then left unattended, while the natural vegetation regenerates [6,39]. Virtually no agro-chemical inputs (e.g., herbicides and inorganic fertilizers) are used in shifting cultivation systems in Myanmar [40]. The most important shifting cultivation crop is upland rice, but maize, pumpkin, beans and pulses are also important (depending on the area) [36].

Estimates of the current extent of shifting cultivation in Myanmar range from about 5 to 20 Mha [41]. In 1993, the Forest Department of Myanmar reported that 22.3% of the country was affected by shifting cultivation, and this number is also still frequently cited [37]. The rotation frequency of shifting cultivation systems in Myanmar varies. In the remote areas of Chin, Bago and Nagaland the fallow periods are up to 10-15 years [36,40]. In other areas—e.g., along the less remote areas of Shan state—the system has been intensified with a reduction of fallow periods to 3–5 years [37].

Shifting cultivation is often considered environmentally destructive, causing deforestation, soil degradation and contributing to $CO_2$ emissions, and policy makers across Southeast Asia—including Myanmar [42]—have tried determinedly to eradicate the system [43]. This notion is, however, increasingly being challenged, as numerous studies have questioned the role of shifting cultivation as a driver of deforestation in MSA and demonstrated the complex system of crops, fallow vegetation and forests to yield various ecosystem services and resources for the benefit of local livelihoods and forest environments in the uplands [22]. For example, shifting cultivation has been shown to maintain positive hydrological properties [5] compared to intensified land uses, reduce soil erosion in ways similar to intact forest in the fallow phase [44–47], and store carbon in quantities that compare to long-rotation perennial plantations [48–50].

Despite the general perception of shifting cultivation as undesirable, the new NLUP acknowledges that lands "*under rotating and shifting cultivation and customary cultivation practices*" should be protected when land is being developed by private entrepreneurs [32]. Furthermore, the policy calls for "*Reclassification, formal recognition and registration of customary land use rights relating to rotating and shifting cultivation that exists in farmland, forestland, vacant land, fallow land, or virgin land shall be recognized in the new National Land Law.*" The fact that 'shifting cultivation' is recognized as a land use category in the NLUP is noteworthy as it is the first time this happens.

### 3.4. Upland Indicator Crops

To provide a quantification of upland agricultural trends, we present the area and production for typical upland agriculture crops based on FAO statistics 1961–2016 (Figure 3), and the potential drivers of these trends for Myanmar. We focus on maize, cassava and rubber because they (i) are dominant crops in upland smallholder systems, (ii) have all seen boom-bust cycles within the past 3 decades, and (iii) are likely to persist as important crops in MSA. We use these three crops as indicators of the suite of boom crops flushing through mainland Southeast Asia, which also includes, e.g., banana, sugar cane, and coffee [51–54]. We focus on the processes of expansion, contraction, intensification and disintensification to understand upland land use trends in Myanmar.

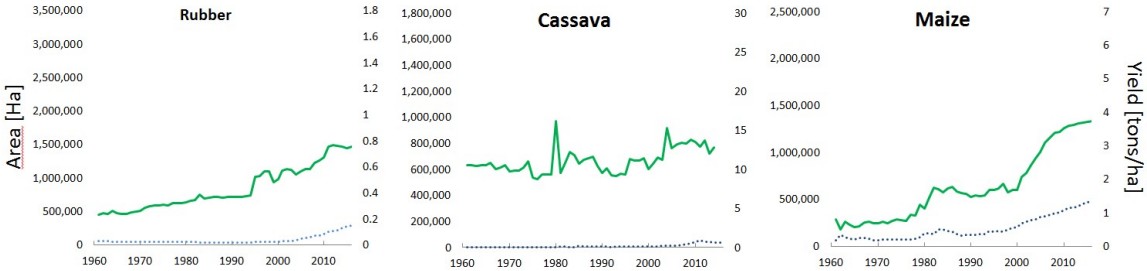

**Figure 3.** Area (blue dots, 1. y-axis) and yield (green solid line, 2. y-axis) of rubber, cassava, and maize for Myanmar 1961–2016.

Myanmar's **rubber** area remained at very modest levels from 1960 until approximately 2005, when liberalization of the rubber trade caused a boom in the area devoted to this crop [55,56], partly financed by the Chinese Opium Replacement Program through large-scale plantations in Shan State [34]. During the same period, yields increased gradually with acceleration after 1995. By 2015, production did not see the burst witnessed in the remaining MSA. Despite sagging market prices, relatively low yields and inferior quality, Myanmar promotes rubber production, processing and exports to the Asian market [57,58]. This strategy is partly based on visions of increasing rubber prices and reduced production from neighboring countries such as Malaysia and Thailand, combined with efforts to close the rubber yield gap in Myanmar by improved breeding and increased farming efficiencies [55].

**Cassava** is mainly grown by smallholders in Myanmar's lowlands, but holds potential as an upland crop [59]. Cassava was virtually not grown in Myanmar until around year 2000. Since then, the area has increased but remains minimal. The yield trend does not reveal any intensification efforts. Between 2012 and 2015, the cassava area declined roughly 30% while yields declined about 14%, allegedly related to a combination of declining prices, government policies, and drought [60,61].

**Maize** was almost absent in the Myanmar area statistics until 1978, followed by a local boom-bust cycle until around 1990. Maize yields mirror this trend. The boom was associated with a long-term hybrid seed research program under the Department of Agricultural Research (DAR) running from 1974. However, the hybrid varieties provided by CIMMYT did not thrive in Myanmar's agro-climatic setting until 1990, when a new, commercial variety was successfully introduced [62]. From 1997 to 2015, the maize area tripled, while yields more than doubled, indicating an intensification process driven by demand from especially China and the Thai CP.

## 4. Land Use Transitions and Driving Forces in Mainland Southeast Asia

In the remaining upland MSA, substantial changes have and are still taking place. As for Myanmar, we present here a synthesis of change processes and their driving forces for Cambodia, Laos, Thailand and Vietnam in order to identify transition pathways which could potentially be replicated in Myanmar. This will be analyzed in the Section 6. Because crop statistics are reported at national level and drivers often apply to an entire country (in contrast to specific regions within the country), the synthesis draws on nation-level statistics and drivers. A more elaborate presentation of land change processes and their drivers for the respective countries is found in the Supplementary Material to this study.

The areas devoted to maize, cassava, and rubber in Mainland Southeast Asia have generally increased since 1960, but this trend covers substantial variation between crops and countries in the timing and expansion and contraction rates (Figure 4). For all three crops, a regional spatial diffusion appears with initial seeding in Thailand, followed later by Vietnam. Laos and Cambodia were the latest adopters, with crops booming after 2000, as has been the case in Myanmar for maize and rubber, while a cassava boom is still pending. For maize and cassava, Thailand saw an area contraction approximately two to three decades after the initial take-off—a pattern also evident for cassava in Vietnam. The area contractions were paralleled by stagnant yield levels. In Vietnam, maize has gained renewed interest since 1990, and both area and yield have increased substantially. A similar trend is observed for cassava since 2000 for Vietnam and Thailand, with Cambodia picking up cassava in 2005 and Maize becoming more popular in both Laos and Cambodia since 2000.

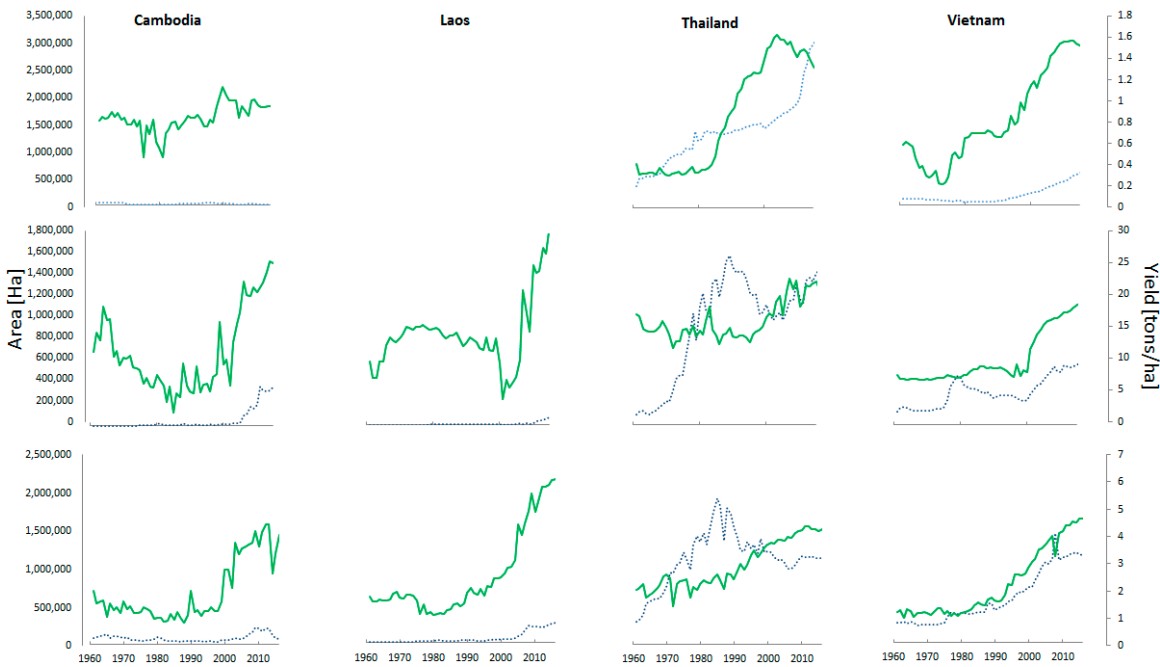

**Figure 4.** Area (blue dots, 1. y-axis) and yield (green solid line, 2. y-axis) of rubber (top row), cassava, and maize (bottom row) 1961–2016 for Cambodia, Laos, Thailand, and Vietnam. Source: FaoStat [ref. 70].

The above-described pattern is a typical example of the introduction of new crops in an area. Early adopters introduce a crop variety possibly not suited to local conditions and farmers are unfamiliar with cultivation techniques and soil suitability. After some time, farmers abandon the crop because of disappointing yields. This phase is followed by government-funded breeding of more suitable varieties, extension services to increase knowledge of cultivation techniques, and possibly subsidies for HYV seeds and inputs, which leads to area expansion and output intensification (Figure 5).

While Myanmar mimicked the contraction/stagnation/intensification trend for maize from around 1975 to present, the late-adopting countries (Cambodia and Laos) leapfrogged the contraction/stagnation phase and are currently following an intensification pathway. For Myanmar, cassava expansion and intensification remain to be seen. In Vietnam, rubber has followed a development similar to the crops outlined above, whereas Thailand hasn't experienced contraction and declining rubber yields. On the contrary, the rubber area expanded continuously, and after a period of stagnant yields, they too exploded in the two decades between 1985 and 2005. In Myanmar, rubber yield improvements began in the mid-1990s, and expansion has only taken off since 2000, and even then, only at modest rates.

To the best of our knowledge, no studies have systematically analyzed the drivers of the upland crop trends for Southeast Asia during the past 3–4 decades described here. While such an effort is also beyond the scope of this study, we provide a brief summary of the economic, political and technological drivers behind the observed trends. Often, political drivers will have an impact on economic drivers, as exemplified below.

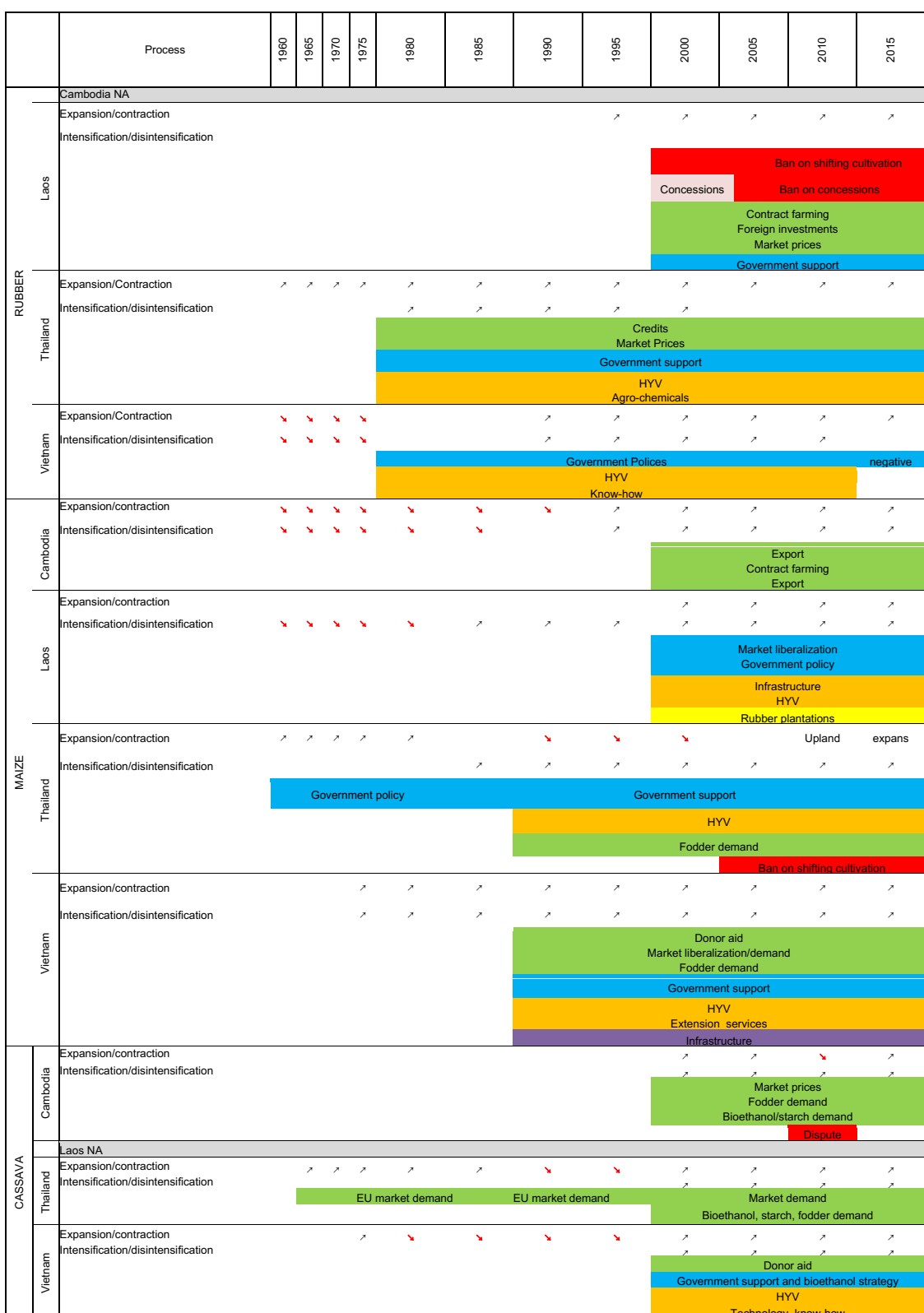

**Figure 5.** Agricultural change processes (expansion, contraction, intensification, and disintensification) and associated drivers 1960–2015 for the three indicator crops (rubber, maize, cassava) in Cambodia, Laos, Thailand and Vietnam. Driver color coding: Orange = technology; Green = economy; Blue = Government policies; Red = ban/dispute; Purple = infrastructure. Bright yellow indicates that maize/rubber cultures were practiced in young rubber plantations in Laos. Arrows indicate the direction of land use processes.

*4.1. Political and Economic Drivers*

The political drivers operate at national levels, at levels specific to the entire agricultural sector, and at crop-specific levels. In MSA, fundamental national political reforms include the transitions from planned economies to market economies and opening to international trade. These include the *Doi Moi* in Vietnam (1986), the New Economic Mechanism in Laos (1986), and the gradual transformation of Cambodia from the socialist, closed, and planned economy in the People's Republic of Kampuchea (1979–1989) through the State of Cambodia (1989–1993) to the Kingdom of Cambodia (1993), now an open market economy [63]. With these reforms came access to regional markets followed by world market access. Thailand had already joined ASEAN in 1967, whereas the remaining countries were latecomers (Vietnam, Laos and Myanmar in 1997, Cambodia in 1999) [64]. As a further means to this end, The ASEAN economic community (AEC), established in 2015, seeks to lift all trade barriers within the group, thereby securing a free flow of commodities [65]. However, Cambodia, Laos, and Myanmar have been granted extensions due to the need for further reforms, and are scheduled to join in 2019.

This market access has allowed producers to respond to market demands and changing market prices. In the early phase of the study period, demand primarily originated from overseas markets, as exemplified by Thai cassava production for the EU fodder market. More recently, Japan and China have emerged as large regional markets (with Chinese rubber demand being the strongest case), and national markets have emerged for, e.g., maize and cassava as feedstocks for the national poultry and fuel industries. The opening of the economies provided foreign companies access to national markets and land. An example of this has been a rush for land for rubber, cassava and banana plantations, as documented thoroughly for Laos and Cambodia, and although with restrictions, Chinese, Vietnamese and Thai companies have negotiated their way to local land possession [66]. However, trade and land access have been subject to import/export subsidies, tariffs and non-tariff trade barriers as well as moratoriums on granting of land concessions (as seen with the volatility in cassava trade across the Laotian/Thai border and Chinese-owned banana plantations in Northern Laos.

Along with these changes came political granting of private land rights such as usufruct rights, leases, or ownership, and rights for private individuals and companies to trade products internationally. These reforms were intended to stimulate the entire economy and incentivized production of boom crops which were in high demand regionally.

However, these policies have also been used to promote intensive commercial cropping at the expense of traditional land use systems. For example, land reforms in Vietnam in 1993 allowed private use rights to land, but also permitted the state to designate areas according to land use type—effectively a ban on upland shifting cultivation. Adverse effects of land rights on smallholders have also been observed in Cambodia [67]. As reported above, large land concessions have been granted to domestic and foreign businesses in Myanmar, bypassing individual land right claims of smallholder farmers.

One of the prime examples of how policies can be a driver of agricultural development is from Thailand, where former Thai president Thaksin in 2001 implemented an extensive set of policies and agricultural credit programs as an attempt to accelerate Thailand's recovery from the Asian crisis by stimulating agricultural growth. In addition, governmental support for development and promotion of High Yielding Varieties (HYVs), sometimes coupled with micro-credits, is a crop-specific political driver. Specific examples include maize subsidies, extension services and distribution of HYVs in Thailand and Vietnam, as well as rubber in Vietnam.

The response of agricultural systems to political and economic drivers is evident in the boom(/bust) cycles of the three indicator crops. A prime example is the recent rubber boom/bust: Soaring increases in area and production [68] was followed by the 2011/2012 bust in rubber prices caused by a combination of reduced demand due to China's economic slow-down, saturation of the regional rubber market and a substantial decrease in crude oil prices, reducing the production costs of synthetic rubber. The price drop was immediately followed by a decrease in production, while area did not contract as swiftly, due to the semi-permanent nature of tree crops.

*4.2. Technological Drivers*

Mechanization, application of agro-chemicals and adoption of HYVs constitute the main technological drivers of agrarian change, while expansion of physical infrastructure contributes to both agricultural and generic rural development. Unfortunately, statistics on the use of fertilizers, pesticides, and tractors remain scarce to date, rendering consistent documentation of changes in technological drivers difficult. Similar lacunae exist for data on road development, although local case studies from Laos have identified expansion of 'feeder' roads to agricultural production areas as important drivers of maize expansion [54,69]. Despite these data paucities, it is beyond question that the road network has expanded throughout Southeast Asia, that use of chemical inputs in agriculture has increased hand-in-hand with the introduction of HYVs, and that mechanization and motorization are ongoing processes. According to FAO, Thailand and Vietnam have been first-developers of agricultural nitrogen use in MSA, with Cambodia and Myanmar now following similar pathways [70].

Sporadic documentation of mechanization confirms this tendency; already by 1983, the agrarian Northeast Thailand held 40,000 2-wheel tractors, a number that had exploded to 1,250,000 by 2003, strongly supported by the government [71]. For Cambodia, agricultural machinery increased 600% from 2001 to 2010, both for small tillers and larger tractors, followed by a doubling of small tillers from 2010 to 2012 [72]. This mechanization has been supported by lifting taxes and import tariffs on machinery [73]. In Myanmar, motorized hand tillers increased 6-fold [74] from 1995 to 2010. These trends are indicators for a general mechanization of agriculture, and uptake rates in MSA's uplands are likely to have lower uptake rates due to sloping terrain.

## 5. Socio-Economic and Environmental Impacts of Recent Land Use Transitions

The socio-economic and environmental implications of the land use transitions outlined above are multi-fold, albeit scarcely documented. In the following, we provide a summary of the current knowledge of the trade-offs between socio-economic and environmental gains and losses of a cash crop transition for upland smallholders.

Tenure, commodification and contract farming are central processes and arrangements for future land use changes, with trade-offs and synergies related to access to land and non-timber forest products, income portfolio diversification, income stability, food security, changes in production and productivity, soil erosion, and carbon storage. Recent reviews document that the environmental and socio-economic effects of transitions from traditional shifting cultivation to intensified land uses in Southeast Asia have predominantly been negative, albeit with a regional and system-specific dependency [19,22].

*5.1. Environmental Impacts of Annual Cropping*

Reported environmental impacts of transitions from shifting cultivation to annual cropping are overwhelmingly negative, and include declining soil quality [75], increased erosion [5], decreased carbon storage [4], reduced biodiversity [7], and pollution from increased use of agro-chemicals [76]. Below, we report some of the environmental impacts associated with the three indicator crops.

5.1.1. Maize

Inappropriate use of pesticides in intensified smallholder maize production systems is a cause of concern in all of MSA [77,78]. Inappropriate use of pesticides in upland areas will likely impact human health through handling and consumption, and indirectly by pollution of water sources, but no direct evidence of links between this and maize cultivation has been found. However, indirect evidence in terms of 'unacceptable' levels of pesticide residues found in blood samples of about half of the tested inhabitants in areas with abundant maize production in Laos indicates serious problems [79]. Concerns about excessive use of pesticides have resulted in a complete ban on the use of pesticides in Hua Pan Province in Laos.

Continuous intensive cultivation of maize promotes high rates of erosion [63] and decreasing soil quality after several years of maize cultivation are often reported by farmers [76,79]. The negative relation between maize cultivation intensity and soil quality has also been documented by studies using bio-chemical indicators [76,80,81]. In Thailand, the negative impacts of conversion from traditional agriculture to intensive production of maize is a cause of public concern and consequently policies are currently being put in place to stop maize cultivation in upland areas [76]. The concerns are mainly related to tree cover loss which is believed to increase the risk of downstream flooding [81]. Scientific evidence of the links between maize expansion, tree cover loss, and increased risk of flooding of other parts of Thailand are, however, absent. Paradoxically, the former ban on shifting cultivation and promotion of annual cropping has resulted in exactly that tree cover loss now feared.

### 5.1.2. Cassava

Cassava has a reputation for depleting soil nutrient reserves and causing degradation of long-term soil quality. This is somewhat related to the confounding fact that cassava is often grown on inherently infertile and acidic soils where other crops would not grow well, and that the use of fertilizers in cassava production systems is often very limited [82]. When cassava is grown for root production and post-harvest residues left on the field, the nutrient removal is lower than for most other crops—including maize [83]. The use of pesticides in cassava production systems is generally low (much lower than in maize production systems). However, cassava is often grown in fragile environments such as the upper parts of hillsides, and production of cassava on slopes has been shown to cause more erosion than production of other crops grown under the same circumstances [82]. This is mainly due to the fact that cassava is planted at a relatively wide spacing and that the initial canopy formation is slow, leaving a large area of soil exposed to the direct impact of rainfall during the first months after planting. The soil disturbance that takes place during harvesting of the roots is also contributing to increased risk of erosion, which is likely to be the most important environmental impact of cassava production.

### 5.2. Environmental Impacts of Perennial Cropping

#### Rubber

The environmental effects of transitions to perennial crop plantations in uplands are ambiguous. The outcome mainly depends on the specific transition pathway, e.g., the system being changed, the type of plantation (e.g., large scale or small scale), the establishment method, and the applied management practices [6].

Studies on the effects of rubber plantations on soil quality are inconclusive. Tanaka et al. [84] reported soil quality in extensively managed rubber plantations to be similar to that of secondary forests, while other studies have reported significant declines of soil organic carbon (SOC) stocks in extensively managed rubber plantations compared to shifting cultivation [48,85]. Evidence from transitions from secondary forests to intensively managed large-scale plantations with mechanical terracing and high amounts of inputs is more conclusive, and points to a decline in soil quality mainly due to large SOC losses and deterioration of soil physical properties [86,87].

Establishment of rubber monocultures on slopes may cause accelerated erosion [5,45]. When the soil surface layer is eroded, subsoils with lower water-holding and infiltration capacity are exposed, which further increase surface run-off and affects the regional hydrology [14]. Moreover, in the dry season, more water is lost by evapotranspiration from rubber plantations than from the mixed-crop cultivated landscapes, potentially negatively affecting groundwater dynamics [88,89]. The high water use of rubber has raised concerns about potential negative effects of continued expansion of rubber plantations on water and food security in upland Southeast Asia [90]. In addition, large-scale rubber plantations are normally established by means of bulldozers, while smallholders rely on hoes, having a comparable less severe effect on erosion compared to large-scale plantations.

The expansion of rubber plantations in Mainland Southeast Asia has to a large extent taken place in biodiversity-rich areas [26,91]. As rubber in this region is almost exclusively grown as monocultures, this has led to substantial biodiversity losses and the transition from shifting cultivation to rubber plantations also reduce landscape and agricultural system diversity and with this, the availability of habitat types for species originating from natural forest [92].

Transitions from short-rotation shifting cultivation to rubber plantations may increase tree cover and carbon storage [4,49], while transition from long-rotation shifting cultivation or areas under forest to rubber plantations causes a net decrease in carbon storage [4,48]. Studies from Xishuangbanna and Laos indicate that the aboveground carbon stock in intensively managed rubber plantations is higher than in low-intensively managed smallholder plantations, which, among other things, is due to a higher stocking density [48,50,93].

## 5.3. Socio-Economic Impacts of Annual And Perennial Crops

Many Southeast Asian countries have experienced economic growth and poverty reduction through agricultural intensification and development of commercial crop production. At the household level, agriculture commercialization has often helped enhance productivity and increase farm income and consumption. However, a range of adverse socio-economic impacts are also typically associated with transitions from traditional agricultural systems to cash cropping [15], and evidence for a negative association between intensification and food security has been documented [94].

Some of the most frequently reported impacts include narrowing of livelihood options, vulnerability to indebtedness, increased labor demand, declining supplies of staple food crops, a reduction in customary practices and sociocultural wellbeing and marginalization of certain groups [95]. Meanwhile, the increase in overall household income may offset the decline in food crop production, the loss of diverse economic activities poses a potential risk to livelihood security and leave the farmers vulnerable to fluctuating yields and prices [15,96].

Smallholders often have to take out loans to facilitate a conversion to intensified cash crop production, which may leave them vulnerable to indebtedness [34,96], as documented for upland maize farmers in Thailand and Laos who are normally relying on seasonal loans to finance the inputs required for maize cultivation. However, fluctuations in yields and grain prices (and in some cases a lack of understanding of the financial system) means that many maize farmers find themselves trapped in vicious spirals of accumulating debt [54,75,79,97].

There are many examples of agricultural commercialization leading to growing land and income inequality among households [98], e.g., in areas of Cambodia that experienced a boom in commercial production of cassava starting in the early 2000s; while cassava in general was a highly profitable crop that brought increased farm income and better capacity to cope with risks, the positive outcomes were skewed in favor of large and medium landholders. Many smallholders relying on loans had to sell land because of limited capacity to cope with farm and household shocks. Hence the transition to intensive production of cassava resulted in growing inequality and contrasting livelihood trajectories [95].

For rubber cultivation, the literature documents substantial differences between the socioeconomic impacts of smallholder production systems and large-scale concessions. Rubber cultivation can be a viable and effective means to generate income for smallholders in the uplands of Southeast Asia and production of rubber by smallholders have been promoted by governments throughout the region [68,76,99]. However, the high investment costs and the monocultural and perennial nature of rubber systems leave smallholders in a lock-in with very limited land-based options for responding to price fluctuations. This became evident with the collapse of the latex prices in 2014—for example, in Northern Laos, where many smallholders who had converted their land to rubber were forced to sell their land to investors. However, as opposed to many other perennial systems, rubber plantations can be left unmanaged for long periods of time, and labor allocated to more profitable activities while waiting for prices to increase [100].

Production of rubber in large-scale concessions increased during the surge of latex prices in the early 2000s [101]. This was especially the case in Laos, where documentation of negative socioeconomic effects of rubber concessions triggered a moratorium on land use concessions that (among other things) halted new concessions for rubber [102]. A large-scale inventory of the quality of land concessions in Laos documented that the majority of rubber concessions (80%) were perceived as predominantly negative by the local population [66]. The most common negatively perceived impacts were the loss of land for farming and livestock, loss of access to non-timber forest products (NTFPs), rising conflicts over land and concerns about the negative effects of agro-chemicals [102]. The most frequently mentioned positive impacts included increased household income, employment opportunities and improved road infrastructure [101].

## 6. Scenarios for Future Pathways

The scenarios outlined here will point to possible future pathways of agrarian change in Myanmar's uplands, guided by Myanmar's recent development and the development in other parts of MSA. We structure the scenarios according to the direction and speed of change along an intensification gradient and discuss the potential roles of and implications for tenureship, actor types, and the environment.

### 6.1. Business as Usual

This scenario follows the pathway laid out by Thailand and Vietnam decades ago. It is characterized by a slow trial-and-error expansion and intensification process where certain crop varieties are promoted with government support, concurrent with government-funded research and development in improved varieties.

The scenario results in modest expansion and yield improvements, or even an initial yield stagnation and area contraction due to (i) the promoted varieties may be unfit for local agro-climatic conditions, (ii) upland areas are sparsely connected to road infrastructure and hence with limited market access, and (iii) local farmers are unfamiliar with new cultivation techniques. The stagnation/contraction phase is followed by years of research and development in improved varieties, expansion of rural road networks and access to local/regional markets, a new introduction of HYVs combined with agricultural extension services and possibly subsidies for agro-chemicals proves more successful, in particular on yield performance.

The main actors in the scenario will be smallholders, agro-businesses, and the Myanmar Government. Smallholders could benefit from government crop campaigns in transitioning from partly subsistence and traditional agriculture to commodity crops. Smallholder uptake of new crops and intensive cultivation methods could be hampered if land titling is not implemented through strong institutions and enforced by just local authorities. Paradoxically, legal land rights could also result in the loss of land, as impoverished or indebted land owners sell their newly recognized land assets, eventually causing a concentration of land on the hand of agro-businesses or large private landowners.

If agro-businesses spearhead this scenario instead of the Myanmar government, there is a risk that smallholders' land will be confiscated and handed to investors, leaving the local population without access to natural resources. In even more severe cases, smallholders have also been forced to work as tenants on the seized land, effectively introducing a feudal land management regime. A third alternative is confiscation of land that is leased back to smallholders, with the risk of indebtedness when the smallholders cannot pay the lease.

The environmental impacts of the slow growth scenario could be of low intensity due to lacking financial backup by agro-businesses, but widespread due to the promoting role of government authorities. We anticipate increased tillage, low application rates of agro-chemicals in annual cropping systems, resulting in loss of soil quality but also moderate leakage of nutrients to the ambient environment, and increasing erosion.

*6.2. Stagnation*

This scenario is a gloomy version of the business-as-usual scenario prescribing little progress for upland smallholders in the coming decade. Several indicators point in this gloomy direction:

(a) Ongoing land grabbing, including land confiscations by the military and the authorities [103], undermine attempts to establish land rights. This is enforced by lack of cross-compliance between land-related policies, as seen with the discrepancies between the National Land Use Plan, the Agriculture Development Strategy, the Farmland Law and the Vacant, Fallow and Virgin Land Law.

(b) Improvements of upland livelihoods are neglected as the government focuses on intensification and maximization of rice, beans, pulses, rubber, and oil palm in the dry zone, the delta zone, and lowlands bordering China.

(c) Potential (foreign) investors in upland areas are repelled by ongoing civil unrest and armed conflicts.

(d) The hopes for democratization with the 2015 election of Aung San Suu Kyi as state counselor are fading with recent setbacks in Myanmar's democratization process. This could negatively influence foreign direct investments and Western tourism.

(e) Slow expansion of the sparse rural road network hinders market integration and limits access to agricultural inputs. Smallholders rely on few market outlets for their crops, which can cause high price volatility.

(f) Farms, fields, and roads in rural uplands are exposed to landslides in the monsoon season, with fatal consequence for smallholders, and loss of crops, landesque capital and roads. These risks provide little incentive to invest in agricultural expansion and intensification in such areas. The Government of Myanmar serves as a main actor in this scenario, primarily by maintaining the status quo, thus allowing persons and companies in positions of power to grab land at the expense of smallholders.

The socio-ecological impacts of this scenario mimic those of the business-as-usual scenario, but with minimal influx of capital to upland economies and lower intensification rates.

*6.3. The Intensification Leapfrogger*

While maize and rubber have been cultivated in Myanmar for decades, yields are comparable to those of Cambodia and inferior to Thailand and Vietnam. Furthermore, the area devoted to the three indicator crops remains low, in particular for cassava, which has been a boom crop in Cambodia, Thailand, and Vietnam, seeing expansion and rapid yield increases.

If the high-yielding varieties grown in the neighboring countries are suited for the agro-climatic conditions in Myanmar, the country could potentially leapfrog into intensive production. However, one or more of the following conditions need to be met for this to happen: (a) granting of land rights and large-scale concessions by Myanmar authorities to domestic or foreign investors; (b) contract farming with links between smallholders and agro-businesses being mediated by brokers, farmers' organizations or cooperatives; (c) implementation of government-lead intensification schemes.

Furthermore, the political, social, and physical environment in a given region should be attractive to investors and agro-businesses, and for lasting success, smallholders should be properly trained to handle new crops and cultivation methods. These requirements would in turn depend on (a) proper infrastructure to secure the access of smallholders to HYV planting material and access to markets, (b) government schemes for agricultural extension services to train farmers, (c) subsidies for agro-chemical inputs to meet regional yield levels, and (d) access to credits through credit institutions. As witnessed in Thailand and Laos, such conditions are sometimes established by agro-businesses, bypassing the role of governments [104].

The intensification trajectory laid out here cannot realistically be applied wall-to-wall across Myanmar by the government. Instead, it is likely to be a hot spot phenomenon, with transitions happening in selected areas and driven by investors and agro-businesses in large-scale concessions, potentially assisted by government-facilitated access to credit for smallholders. The environmental

and socio-economic impacts will depend on these modes of deployment and could lead to socio-economic differentiation between provinces with/without investments, between villages and between households in the villages as described for Cambodia above. A transition towards mono-cropping could expose local communities to market price volatilities and impact food provisioning negatively, especially if the cultivar is not part of the traditional diet. Furthermore, intensification is labor demanding and would require either local labor availability or construction of infrastructure to bring labor and/or machinery into the affected areas. Compared to the business as usual scenario, the environmental impacts of the intensification leapfrogger scenario could be intense but limited to transition hot spots.

The positive factors of intensification could be capacity building regarding cultivation methods among local households, influx of cash in the local economy, and increased accessibility due to road constructions.

The environmental impacts follow those described in the *environmental impacts* section, including changing soil quality, erosion, agro-chemical leakage, and depending on crop type, changes to tree cover, carbon storage, biodiversity, etc.

Currently, concessions and contract farming projects are ongoing in Shan State along the Thai and Chinese borders (maize) and Southern Myanmar (rubber, oil palm), whereas Western Myanmar along the Indian and Bangladeshi borders receive little attention from investors.

*6.4. The Sustainable Leapfrogger*

Agricultural intensification has been identified as a way to feed a growing population and reserving land areas for ecosystem conservation (ref support of land sparing), but agricultural intensification often carries negative socio-economic and environmental impacts, including environmental degradation and lack of local food security and livelihood improvements. As a way forward, sustainable intensification is a process benefitting all components of the socio-ecological system. This includes conserving or improving bundles of ecosystem services, increasing production of food and fibers while securing access rights to land and food for various groups, including vulnerable people. Due to Myanmar's relatively large forest area and low-intensive agriculture, a potential option exists to skip the well-known agricultural development pathways with negative socio-ecological outcomes and leapfrog to a sustainable solution. The prerequisites of a sustainable intensification pathway would be (a) official recognition of the socio-economic and environmental tradeoffs involved in conventional development pathways, (b) willingness to balance production, socio-economy, and the environment, (c) securing land titles for rural smallholders, including communal/customary and private rights, (b) focus on holistic socio-ecological resilience, in contrast to crop production maximization.

These prerequisites are recognized in Myanmar's Land Use Plan (the Republic of the Union of Myanmar 2016) and partly in the Agricultural Development Strategy (Myanmar 2018). However, contradictorily, the latter also has a strong focus on production maximization, and the existing Farmland Law and the Vacant, Fallow and Virgin Land Law allows granting of land to investors, including areas confiscated from rural communities.

The main actor driving sustainable intensification in Myanmar should be the Republic itself, by establishing and pursuing clear, univocal plans and policies, strong institutions to regulate land use, and local enforcement of institutional rules. Smallholders, in particular in marginal uplands under customary/communal land management, would be the main beneficiaries, e.g., if the acknowledgement of socio-ecological co-benefits of traditional farming systems could manifest as payments for ecosystem services. Such efforts would be backed up by NGOs and state donors.

The environmental impacts of this scenario would be limited, as land use transitions should move along a sustainable pathway. This could include promotion of land sharing, maintaining partial tree cover on farm fields, reduced or no-tillage, and utilization and recycling of organic fertilizers available in the land use systems, e.g., from foliage, and integrated livestock and aquaculture.

## 7. Conclusions

Boom crops have swept across mainland Southeast Asia in the past decades, along with land use transitions and livelihood changes. Due to its particular history and political regime, Myanmar is at the tail of these transitions. However, international economic interests have entered Myanmar and brought both potentials and, perhaps, perils to upland smallholders at the margins of society, with consequences for local ecosystems and livelihoods. In the absence of an orbuculum, we cannot predict how the future plays out in Myanmar, but the pathways taken by other countries of Mainland Southeast Asia bring important lessons which could assist clever land use strategies and policies that integrate environmental and social dimensions of rural and economic transitions to minimize socio-ecological tradeoffs.

However, Myanmar is challenged by a strong regime with weak democratic institutions. This is evident in the lack of land rights, in particular, recognition of traditional communal tenure systems, and in the brute land grabs seizing smallholder land for concessions and development. A gap remains between the reforms recognizing customary/communal land rights in the National Land Use Policy, and the ongoing land grabbing, effectively pointing to a regressive land reform. In addition, upland smallholder farming systems are neglected in the agricultural development plans which focus on production of rice, beans and pulses in lowlands.

The question, then, remains: what awaits upland farmers of Myanmar? Among the neighboring MSA countries, Myanmar shares traits with Laos and Cambodia in being late adopters of the boom crops rubber, maize, and cassava. Like Laos, Myanmar is also prone to land grabbing and granting of concessions to Chinese and Thai investors, and the three countries represent MSA's poorest populations. It would thus not be surprising if Myanmar mimics the recent upland developments in Laos. This points to the intensification leapfrogger scenario with sporadic intensification in hot spot areas while the vast majority of uplands are left in the slow growth scenario. The recent critical stances of Western governments and the UN and the resulting unwillingness to invest in Myanmar could indicate a slowdown for Myanmar, making the stagnation scenario probable.

All authors contributed equally to conceptualization, data curation, writing and editing this manuscript. We declare no conflicts of interest. We thank two anonymous reviewers for constructive comments which assisted improving the manuscript.

**Supplementary Materials:** The following are available online at http://www.mdpi.com/2073-445X/8/2/29/s1. References [105–125] are cited in the supplementary materials.

**Author Contributions:** Conceptualization, M.R.J., M.P., T.B.B.; methodology, M.R.J., M.P., T.B.B.; software, N/A.; validation, N/A.; formal analysis, M.R.J., M.P., T.B.B.; investigation, M.R.J., M.P., T.B.B.; resources, N/A.; data curation, M.R.J., M.P., T.B.B..; writing—original draft preparation, M.R.J., M.P., T.B.B.; writing—review and editing, M.R.J., M.P., T.B.B.; visualization, M.R.J., M.P., T.B.B.; supervision, N/A.; project administration, N/A; funding acquisition, N/A.

**Funding:** This research received no external funding.

**Acknowledgments:** We thank two anonymous reviewers for constructive comments which assisted improving the manuscript.

**Conflicts of Interest:** The authors declare no conflict of interest.

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
