# Peer review of "What Awaits Myanmar’s Uplands Farmers? Lessons Learned from Mainland Southeast Asia"

_land, doi:10.3390/land8020029_

Round 1

Reviewer 1 Report

The paper presents kind of scientific literature and report review on land use changes in the region of Southeast Asia with a focus on Myanmar. It is well prepared, based on extensive literature and very interesting. However, I have some suggestions, which might help the Authors to slightly improve it. Please find them below.

-          The paper suggest that it is focused on Myanmar’s uplands, while most of the work is on Mainland Southeast Asia in general (even not on the upland areas in the neighboring countries) – can we compare the data for whole countries in the region (as presented e.g. on Figure 3) to the specific uplands context of Myanmar? Could you clarify on that?

-          The upland itself is defined as lands with slopes steeper than 15 degrees – shouldn’t ‘uplands’ be also reflected by elevation?

-          The Figures needs rethinking e.g.:

o   Figure 2 – wouldn’t it be better to have similar scale on Y axis for all three crops to make it comparable?

o   It is very hard to read Figure 4 – please redesign it

-          The technical part of the manuscript needs to be improved:

o   There are citing style problems in lines 371, 384, 394 –– e.g. [81]101

o   L 96 – “(…) land has never been developed and is still used farmed by the original users [24].” – should be either used or farmed, I think

o   In the first line of the Abstract you write “South-east Asia”, but throughout the paper you use “Southeast Asia” – could you please unify?

o   In Supplementary material “Laos”, as the paragraph title should be in Italics

o   The affiliations and final statements and declarations of Authors contibution etc. are not in line with LAND guidelines

-          Much of the work on land use change in the region, e.g. in Vietnam was published by Patrick Meyfroidt, but I cannot find it in the references, although he uses remote sensing techniques, which are helpful, when official data are not easily available.

Author Response

Dear Reviewer,

Thanks for your thoughtful and thought-inspiring comments. We address each comment below:

The paper presents kind of scientific literature and report review on land use changes in the region of Southeast Asia with a focus on Myanmar. It is well prepared, based on extensive literature and very interesting. However, I have some suggestions, which might help the Authors to slightly improve it. Please find them below.

Thanks for the kind words.

-          The paper suggest that it is focused on Myanmar’s uplands, while most of the work is on Mainland Southeast Asia in general (even not on the upland areas in the neighboring countries) – can we compare the data for whole countries in the region (as presented e.g. on Figure 3) to the specific uplands context of Myanmar? Could you clarify on that?

This is a good point. We seek to understand the situation and potential future for Myanmar’s upland smallholders in light of the development in the remaining Mainland Southeast Asia (MSA), hence the attention paid to the region. For this reason, there is clearly a scale mismatch (uplands versus nations). Unfortunately, this is difficult to avoid for two reasons: a) our focus and concern is on traditional smallholder systems in Myanmar. These are mostly located in uplands (hence the upland focus).  b) information and statistics on the agricultural development in Myanmar and MSA is not disaggregated on uplands/lowlands. Rather, they are reported at national level, and many of the policies and drivers we describe operate at national level, not specifically targeted uplands. Therefore, we focus on the three indicator crops which are typically found in upland smallholder systems (“To provide a quantification of upland agricultural trends, we present the area and production for typical upland agriculture crops based on FAO statistics 1961-2016 (Figure 2), and the potential drivers of these trends for Myanmar. We focus on maize, cassava and rubber because they i) are dominant crops in upland smallholder systems, ii) have all seen boom-burst cycles within the past 3 decades, and iii) are likely to persist as important crops in MSA”, lines 184-190).

To clarify this, we have added the following sentence (ll. 234-236): “Because crop statistics are reported at national level and drivers often apply to an entire country (in contrast to specific regions within the country), the synthesis draws on nation-level statistics and drivers.”

-          The upland itself is defined as lands with slopes steeper than 15 degrees – shouldn’t ‘uplands’ be also reflected by elevation?

We also wondered about this and searched for a clear definition of uplands/upland agriculture. Since none seems to exist, we decided to define our application of the term clearly. In our case study, elevation is judged to be less important for the choice of farming system than the slope of the land because elevation isn’t a limiting factor for the crops we consider in the specific region, and because slope is associated with farming system relevant choices and constraints like degree of mechanization and management, environmental impacts (notably leakage and soil erosion), and land rights We therefore added this clarification: “This definition excludes e.g. high plains, i.e. relatively level lands at higher altitudes, because these are suitable for mechanized agriculture and thus not representative of traditional smallholder agriculture” (ll. 142-144)

The Figures needs rethinking e.g.:

o   Figure 2 – wouldn’t it be better to have similar scale on Y axis for all three crops to make it comparable?

Thanks for the comment. We had the exact same consideration and also produced figures with similar scales. However, due to the very different areas and yields between crops a uniform scale would disguise some of the trends (e.g. cassava area in Myanmar). Instead we opted for consistency between Figures 2 and 3 (the graphs of Myanmar and MSA, respectively).

o   It is very hard to read Figure 4 – please redesign it

Right. The formatting of Figure was shaken when fitted to the Land template. We have inserted a temporary Figure in the text and refer to the uploaded pdf version.

-          The technical part of the manuscript needs to be improved:

o   There are citing style problems in lines 371, 384, 394 –– e.g. [81]101

Thanks for pointing this out. We have revised the manuscript accordingly.

·      L 96 – “(…) land has never been developed and is still used farmed by the original users [24].” – should be either used or farmed, I think

Thank you for spotting that mistake. We have changed the text to “farmed”

·      In the first line of the Abstract you write “South-east Asia”, but throughout the paper you use “Southeast Asia” – could you please unify?

Done!

·      In Supplementary material “Laos”, as the paragraph title should be in Italics

Thank you. We have changed the text according to your comment.

The affiliations and final statements and declarations of Authors contribution etc. are not in line with LAND guidelines

We have added author affiliations. Regarding the declarations, Land writes:

“For research articles with several authors, a short paragraph specifying their individual contributions must be provided. The following statements should be used "Conceptualization, X.X. and Y.Y.; Methodology, X.X.; Software, X.X.; Validation, X.X., Y.Y. and Z.Z.; Formal Analysis, X.X.; Investigation, X.X.; Resources, X.X.; Data Curation, X.X.; Writing – Original Draft Preparation, X.X.; Writing – Review & Editing, X.X.; Visualization, X.X.; Supervision, X.X.; Project Administration, X.X.; Funding Acquisition, Y.Y.”, please turn to the CRediT taxonomy for the term explanation. For more background on CRediT, see here. "Authorship must include and be limited to those who have contributed substantially to the work. Please read the section concerning the criteria to qualify for authorship carefully".

·  Conflicts of Interest: Authors must identify and declare any personal circumstances”

Since all authors contributed equally, we take the position that it will be waste of text to declare each author individually. Also, our study contains the processes conceptualization, data curation, writing and editing, not e.g. software, validation etc. We declare those processes that our work contains, and our declaration thus reads:

“All authors contributed equally to conceptualization, data curation, writing and editing this manuscript. We declare no conflicts of interest.”

If the editors would like a full declaration, also of the processes we have not undertaken, we will be happy to do so.

-          Much of the work on land use change in the region, e.g. in Vietnam was published by Patrick Meyfroidt, but I cannot find it in the references, although he uses remote sensing techniques, which are helpful, when official data are not easily available.

Thanks for the insight. We are aware of Dr. Meyfroidt’s impressive contribution to land system science and his work on Asia, in particular his work with forest transitions in Vietnam and generally in land transitions. We have not succeeded in finding publications with national/regional focus on upland developments or drivers of agricultural change processes in MSA. Any suggestions to publications we have missed would be highly appreciated.

Reviewer 2 Report

This paper examines the agricultural transition currently taking place in Myanmar. This paper is well-written and generally well-organized. There are a few key areas which need to be attended to before this paper would be ready for publication in this journal or a journal of similar quality.

To start, the methods need to be more clearly delineated. The authors do not have a methods section, nor is it clear what they actually did methodologically or epistemologically in terms of developing their argument in the paper. While the authors clearly lay out their research questions in the introduction, it is unclear the methods they undertake to answer these research questions. I suggest a logical flow model or diagram in addition to a description of the methods. In addition, from this gap, it is hard to understand how the authors can make such an intellectual leap as they do at the end of the paper if the reader is unsure how the argument was supported/supported by certain pieces of evidence (from specific methods undertaken which are unclear at the moment). 

Second, while Myanmar is clearly an interesting case, the authors need to more clearly articulate why and how this can or should be related to other countries in the region. What are the strengths and weaknesses of this comparison? Also, can the authors make the analytical leap they do in the end of the paper or is this too speculative given the evidence provided and strengths/weaknesses of choosing the Myanmar case?

Third, the above issues stated thus far also relate to the paper's organization, as the authors could consider rethinking how they construct and organize the paper's different sections, so that it makes more sense analytically and follows a clearer method. This organization would help the reader understand how the conclusions/results are more supported and less speculative following a logical flow method discussed earlier in the paper.

Fourth, while the authors provide great discussion on their topic, there is also an opportunity to position their research more clearly in the existing body of literature on this topic in the field. This would allow the authors to more clearly delineate the significance of their findings, relevance for existing literature, and areas for future research. This is already in the paper to an extent, but could be pushed a bit further.

Good luck with the revisions.

Author Response

Dear Reviewer,

Thanks for taking the time to review our manuscript. We sincerely appreciate your comments.

This paper examines the agricultural transition currently taking place in Myanmar. This paper is well-written and generally well-organized. There are a few key areas which need to be attended to before this paper would be ready for publication in this journal or a journal of similar quality.

We thank the reviewer for the praising words.

To start, the methods need to be more clearly delineated. The authors do not have a methods section, nor is it clear what they actually did methodologically or epistemologically in terms of developing their argument in the paper. While the authors clearly lay out their research questions in the introduction, it is unclear the methods they undertake to answer these research questions. I suggest a logical flow model or diagram in addition to a description of the methods.

Thanks, this is a valuable comment. Indeed, our study does not follow a standard IMRAD convention and fits somewhere among the genres review article, research article, and opinion letter.  Prior to undertaking the study, we scrutinized similar contributions to Land and were unable to identify a common manuscript structure. We thus stick to the structure we have and add a methodology section and a flow diagram:

“Land use, land use changes, and their economic, social, and environmental impacts, have been studied throughout MSA by academics, NGOs and donor agencies. Many academic studies are conducted and published as local-scale studies while there is a scarcity of synthesis studies generalizing local trends and drivers and a lack of national-scale. A large body of knowledge is published as reports, working papers, etc. Finally, Myanmar has only recently been accessible for foreign researchers, further limiting the available knowledge base. These conditions render a systematic review of general land changes and associated impacts difficult. Our methodological approach has therefore been eclectic and pragmatic (Figure 1); to assess the current state of Myanmar’s upland land use systems (RQ 1), we complemented the few published peer reviewed studies with broad Google Scholar and general Internet searches for relevant reports and working papers, in addition to personal experience from field visits in Chin State. Based on these sources, we provide a summary of recent land use reforms and an overview of the traditional upland farming system, shifting cultivation. We used time series statistics from FAOSTAT on three indicator upland crops (rubber, cassava, maize) to describe expansion, contraction, intensification, and disintensification processes and relate those to drivers of change identified in the references. A similar approach was followed to identify the driving forces of land use transitions in MSA (RQ 2)), where we also included newspaper articles as a source. The key social, economic, and environmental impacts of land transitions (RQ 3) where estimated qualitatively by consulting the authors’ own publications and reference databases and discussed individually for the three indicator upland crops. The results of RQs 1, 2, and 3 informs a discussion of potential pathways for future land use transitions in Myanmar (RQ 4) where we point to possible social, economic and environmental consequences of these. The structure of the article follows this progression through the research questions.”

(see attached pdf for Figure)

Figure 1 Logical flow model of the manuscript. Blue boxes refer to research questions, green ellipses to the applied methods and data sources.

In addition, from this gap, it is hard to understand how the authors can make such an intellectual leap as they do at the end of the paper if the reader is unsure how the argument was supported/supported by certain pieces of evidence (from specific methods undertaken which are unclear at the moment). 

Thanks for this comment. We realize that the applied methods and how we arrived at the transition scenarios wasn’t clear in the first manuscript. With the added methods section and flow diagram we hope to have added clarity to this and closed the analytical leap.

Second, while Myanmar is clearly an interesting case, the authors need to more clearly articulate why and how this can or should be related to other countries in the region. What are the strengths and weaknesses of this comparison? Also, can the authors make the analytical leap they do in the end of the paper or is this too speculative given the evidence provided and strengths/weaknesses of choosing the Myanmar case?

Thank you for these constructive comments. We address them individually here:

1)    the authors need to more clearly articulate why and how this can or should be related to other countries in the region. What are the strengths and weaknesses of this comparison?

Add 1: The reviewer is right. Thanks for this constructive comment. We have added a paragraph below our research questions addressing this issue:
We hypothesize that upland agricultural land use changes and impacts in Myanmar are or may become comparable to those seen in the remaining MSA. This is supported by comparable agro-climatic conditions, similarities in traditional farming systems (notably shifting cultivation), joint experiences with totalitarian and/or military government regimes, recent liberalizations of national economic systems, and the exposure to market drivers, megatrends promoting boom crops, contract farming, and a growing regional demand for food, fibers and energy. However, this hypothesis could fail due to Myanmar’s particularities: being the Eastern border post towards India, Myanmar has direct physical access to the Indian market with different dietary preferences than MSA. This could impact crop choice and farming system in Myanmar. Further, despite a ceasefire agreement national hegemony has not been reached, and armed conflicts still occur between government military and armed rebellions. Finally, we can’t predict if the drivers and changes seen in MSA during the past 50 years will repeat, or if new megatrends and boom crops will emerge.

We revisit this hypothesis in the discussion:

“Boom crops have swept across mainland Southeast Asia in the past decades along with land use transitions and livelihood changes. Due to its particular history and political regime, Myanmar is at the tail of these transitions. However, international economic interests have entered Myanmar and brought both potentials and perhaps perils to upland smallholders at the margins of society, with consequences for local ecosystems and livelihoods. In lack of an orbuculum we can't predict how the future plays out in Myanmar, but the pathways taken by the other countries of Mainland Southeast Asia bring important lessons which could assist clever land use strategies and policies that integrate environmental and social dimensions of rural and economic transitions to minimize socio-ecological tradeoffs.

However, Myanmar is challenged by a strong regime with weak democratic institutions. This is evident in the lack of land rights, in particular recognition of traditional communal tenure systems, and in the brute land grabs seizing smallholder land for concessions and development. A gap remains between the reforms recognizing customary/communal land rights in the National Land Use Policy, and the ongoing land grabbing, effectively pointing to a regressive land reform. In addition, upland smallholder farming systems are neglected in the agricultural development plans which focus on production of rice, beans and pulses in lowlands.

The question then remains, what await upland farmers of Myanmar? Among the neighboring MSA countries, Myanmar shares traits with Laos and Cambodia in being late adopters of the boom crops rubber, maize, and cassava. Like Laos, Myanmar is also prone to land grabbing and granting of concessions to Chinese and Thai investors, and the three countries represent MSA’s poorest populations. It would thus not be surprising if Myanmar mimics the recent upland developments in Laos. This points to the intensification leapfrogger scenario with sporadic intensification in hot spot areas while the vast majority of uplands are left in the slow growth scenario. The recent critical stands of Western governments and the UN and resulting unwillingness to invest in Myanmar could indicate a slowdown for Myanmar, making the stagnation scenario probable.”

2)    can the authors make the analytical leap they do in the end of the paper or is this too speculative given the evidence provided and strengths/weaknesses of choosing the Myanmar case?

We hope to have sufficiently clarified our hypothesis, methods, and the logical flow through the paper in our previous comments and thus also addressed this comment.

Third, the above issues stated thus far also relate to the paper's organization, as the authors could consider rethinking how they construct and organize the paper's different sections, so that it makes more sense analytically and follows a clearer method. This organization would help the reader understand how the conclusions/results are more supported and less speculative following a logical flow method discussed earlier in the paper.

We have added a paragraph in our hypotheses and revisit those in the conclusion. We have also added a methodology section and a diagram of the flow through the manuscript (please refer to previous comments). We hope this has added clarity to the paper’s organization.

Fourth, while the authors provide great discussion on their topic, there is also an opportunity to position their research more clearly in the existing body of literature on this topic in the field. This would allow the authors to more clearly delineate the significance of their findings, relevance for existing literature, and areas for future research. This is already in the paper to an extent, but could be pushed a bit further.

We thank the reviewer for this comment. We have tried to cover all peer reviewed studies of land transitions, their drivers and impacts, for mainland Southeast Asia, but might have missed some? All suggestions to relevant literature will be highly appreciated. Due to the scarcity of such studies, we have scavenged the Internet for grey literature, including newspapers, to unveil drivers and impacts which we could relate to the land change processes identified through the FAOstat data. With a total of 125 references, we evaluate that we have read a decent share of relevant written material, but we can’t exclude that we may have missed important contributions. Again, we will happily include such references, should the reviewer have such at hand.

Good luck with the revisions.

Thank you. The comments provided have been very helpful in improving the manuscript.

Round 2

Reviewer 2 Report

The authors have done a good job attending to the reviewers' comments. For this reason, I think this paper is ready to be published, following a review of the paper for readability and clarity of language.

Good work!